# Feedforward Learning of Mixture Models

**Matthew Lawlor**[*]
Applied Math
Yale University
New Haven, CT 06520
mflawlor@gmail.com

**Steven W. Zucker**
Computer Science
Yale University
New Haven, CT 06520
zucker@cs.yale.edu

## Abstract

We develop a biologically-plausible learning rule that provably converges to the class means of general mixture models. This rule generalizes the classical BCM neural rule within a tensor framework, substantially increasing the generality of the learning problem it solves. It achieves this by incorporating triplets of samples from the mixtures, which provides a novel information processing interpretation to spike-timing-dependent plasticity. We provide both proofs of convergence, and a close fit to experimental data on STDP.

## 1 Introduction

Spectral tensor methods and tensor decomposition are emerging themes in machine learning, but they remain global rather than "online." While incremental (online) learning can be useful for applications, it is essential for neurobiology. Error back propagation does operate incrementally, but its neurobiological relevance remains a question for debate. We introduce a triplet learning rule for mixture distributions based on a tensor formulation of the BCM biological learning rule. It is implemented in a feedforward fashion, removing the need for backpropagation of error signals.

The triplet requirement is natural biologically. Informally imagine your eyes microsaccading during a fixation, so that a tiny image fragment is "sampled" repeatedly until the next fixation. Viewed from visual cortex, edge selective neurons will fire repeatedly. Importantly, they exhibit strong statistical dependencies due to the geometry of objects and their relationships in the world. "Hidden" information such as edge curvatures, the presence of textures, and lighting discontinuities all affect the probability distribution of firing rates among orientation selective neurons, leading to complex statistical interdependencies between neurons.

Latent variable models are powerful tools in this context. They formalize the idea that highly coupled random variables can be simply explained by a small number of hidden causes. Conditioned on these causes, the input distribution should be simple. For example, while the joint distribution of edges in a small patch of a scene might be quite complex, the distribution conditioned on the presence of a curved object at a particular location might be comparatively simple [14]. The specific question is whether brains can learn these mixture models, and how.

*Example*: Imagine a stimulus space of $K$ inputs. These could be images of edges at particular orientations, or audio tones at $K$ frequencies. These stimuli are fed into a network of $n$ Linear-Nonlinear Poisson (LNP) spiking neurons. Let $r_{ij}$ denote the firing rate of neuron $i$ to stimulus $j$. Assuming the stimuli are drawn independently with probability $\alpha_k$, then the number of spikes $\boldsymbol{d}$ in an interval where a single stimulus is shown is distributed according to a mixture model.

$$P(\boldsymbol{d}) = \sum_k \alpha_k P_k(\boldsymbol{d})$$

where $P_k(\boldsymbol{d})$ is a vector of independent Poisson distributions, and the rate parameter of the $i$th component is $r_{ik}$. We seek a filter that responds (in expectation) to one and only one stimulus. To do this, we must learn a set of weights that are orthogonal to all but one of the vectors of rates $\boldsymbol{r}_{\cdot j}$. Each rate vector corresponds to the mean of one of the mixtures. Our problem is thus to learn the means of mixtures. We will demonstrate that this can be done non-parametrically over a broad class of firing patterns, not just Poisson spiking neurons.

Although fitting mixture models can be exponentially hard, under a certain multiview assumption, non-parametric estimation of mixture means can be done by tensor decomposition [2][1]. This multiview assumption requires access to at least 3 *independent copies* of the samples; i.e., multiple samples drawn from the same mixture component. For the LNP example above, this multiview assumption requires only that we have access to the number of spikes in three disjoint intervals, while the stimulus remains constant. After these intervals, the stimulus is free to change – in vision, say, after a saccade – after which point another sample triple is taken.

Our main result is that, with a slight modification of classical Bienenstock-Cooper-Munro [5] synaptic update rule a neuron can perform a tensor decomposition of the input data. By incorporating the interactions between input triplets, our online learning rule can provably learn the mixture means under an extremely broad class of mixture distributions and noise models. (The classical BCM learning rule will not converge properly in the presence of noise.) Specifically we show how the classical BCM neuron performs gradient ascent in a tensor objective function, when the data consists of discrete input vectors, and how our modified rule converges when the data are drawn from a general mixture model.

The multiview requirement has an intriguing implication for neuroscience. Since spikes arrive in waves, and spike trains matter for learning [9], our model suggests that *the waves of spikes arriving during adjacent epochs in time provide multiple samples of a given stimulus*. This provides an unusual information processing interpretation for the functional role of spike trains. To realize it fully, we point out that classical BCM can be implemented via spike timing dependent plasticity [17][10][6][18]. However, most of these approaches require much stronger distributional assumptions on the input data (generally Poisson), or learn a much simpler decomposition of the data than our algorithm. Other, Bayesian methods [16], require the computation of a posterior distribution which requires an implausible normalization step. Our learning rule successfully avoids these issues, and has provable guarantees of convergence to the true mixture means. At the end of this paper we show how our rule predicts pair and triple spike timing dependent plasticity data.

## 2 Tensor Notation

Let $\otimes$ denote the tensor product. We denote application of a $k$-tensor to $k$ vectors by $T(\boldsymbol{w}_1, ..., \boldsymbol{w}_k)$, so in the simple case where $T = \boldsymbol{v}_1 \otimes ... \otimes \boldsymbol{v}_k$,

$$T(\boldsymbol{w}_1, ..., \boldsymbol{w}_k) = \prod_j \langle \boldsymbol{v}_j, \boldsymbol{w}_j \rangle$$

We further denote the application of a $k$-tensor to $k$ matrices by $T(M_1, ..., M_k)$ where

$$T(M_1, ..., M_k)_{i_1, ..., i_k} = \sum_{j_1, ..., j_k} T_{j_1, ..., j_k} [M_1]_{j_1, i_1} ... [M_k]_{j_k, i_k}$$

Thus if $T$ is a symmetric 2-tensor, $T(M_1, M_2) = M_1^T T M_2$ with ordinary matrix multiplication. Similarly, $T(\boldsymbol{v}_1, \boldsymbol{v}_2) = \boldsymbol{v}_1^T T \boldsymbol{v}_2$.

We say that $T$ has an orthogonal tensor decomposition if

$$T = \sum_k \alpha_k \boldsymbol{v}_k \otimes \boldsymbol{v}_k \otimes ... \otimes \boldsymbol{v}_k \quad \text{and} \quad \langle \boldsymbol{v}_i, \boldsymbol{v}_j \rangle = \delta_{ij}$$

## 3 Connection Between BCM Neuron and Tensor Decompositions

The BCM learning rule was introduced in 1982 in part to correct failings of the classical Hebbian learning rule [5]. The Hebbian learning rule [11] is one of the simplest and oldest learning rules. It

posits that the selectivity of a neuron to input $i$, $\boldsymbol{m_t}(i)$ is increased in proportion to the post-synaptic activity of that neuron $c_t = \langle \boldsymbol{m_{t-1}}, \boldsymbol{d_t} \rangle$, where $\boldsymbol{m}$ is a vector of synaptic weights.

$$\boldsymbol{m_t} - \boldsymbol{m_{t-1}} = \gamma_t c_t \boldsymbol{d_t}$$

This learning rule will become increasingly correlated with its input. As formulated this rule does not converge for most input, as $\|\boldsymbol{m}\| \to \infty$. In addition, in the presence of multiple inputs Hebbian learning rule will always converge to an "average" of the inputs, rather than becoming selective to one particular input. It is possible to choose a normalization of $\boldsymbol{m}$ such that $\boldsymbol{m}$ will converge to the first eigenvector of the input data. The BCM rule tries to correct for the lack of selectivity, and for the stabilization problems. Like the Hebbian learning rule, it always updates its weights in the direction of the input, however it also has a sliding threshold that controls the magnitude and sign of this weight update.

The original formulation of the BCM rule is as follows: Let $c$ be the post-synaptic firing rate, $\boldsymbol{d} \in \mathbb{R}^N$ be the vector of presynaptic firing rates, and $\boldsymbol{m}$ be the vector of synaptic weights. Then the BCM synaptic modification rule is

$$c = \langle \boldsymbol{m}, \boldsymbol{d} \rangle$$
$$\dot{\boldsymbol{m}} = \phi(c, \theta)\boldsymbol{d}$$

$\phi$ is a non-linear function of the firing rate, and $\theta$ is a sliding threshold that increases as a superlinear function of the average firing rate.

There are many different formulations of the BCM rule. The primary features that are required are $\phi(c, \theta)$ is convex in $c$, $\phi(0, \theta) = 0$, $\phi(\theta, \theta) = 0$, and $\theta$ is a super-linear function of $E[c]$.

These properties guarantee that the BCM learning rule will not grow without bound. There have been many variants of this rule. One of the most theoretically well analyzed variants is the Intrator and Cooper model [12], which has the following form for $\phi$ and $\theta$.

$$\phi(c, \theta) = c(c - \theta) \text{ with } \theta = E[c^2]$$

**Definition 3.1** (BCM Update Rule). With the Intrator and Cooper definition, the BCM rule is defined as

$$\boldsymbol{m_t} = \boldsymbol{m_{t-1}} + \gamma_t c_t (c_t - \theta_{t-1})\boldsymbol{d_t} \tag{1}$$

where $c_t = \langle \boldsymbol{m_{t-1}}, \boldsymbol{d_t} \rangle$ and $\theta = E[c^2]$. $\gamma_t$ is a sequence of positive step sizes with the property that $\sum_t \gamma_t \to \infty$ and $\sum_t \gamma_t^2 < \infty$

The traditional application of this rule is a system where the input $\boldsymbol{d}$ is drawn from linearly independent vectors $\{\boldsymbol{d_1}, ..., \boldsymbol{d_k}\}$ with probabilities $\alpha_1, ..., \alpha_K$, with $K = N$, the dimension of the space.

These choices are quite convenient because they lead to the following objective function formulation of the synaptic update rule.

$$R(\boldsymbol{m}) = \frac{1}{3}E\left[\langle \boldsymbol{m}, \boldsymbol{d} \rangle^3\right] - \frac{1}{4}E\left[\langle \boldsymbol{m}, \boldsymbol{d} \rangle^2\right]^2$$

Thus,

$$\nabla R = E\left[\langle \boldsymbol{m}, \boldsymbol{d} \rangle^2 \boldsymbol{d} - E[\langle \boldsymbol{m}, \boldsymbol{d} \rangle^2]\langle \boldsymbol{m}, \boldsymbol{d} \rangle \boldsymbol{d}\right]$$
$$= E[c(c - \theta)\boldsymbol{d}]$$
$$= E[\phi(c, \theta)\boldsymbol{d}]$$

So in expectation, the BCM rule performs a gradient ascent in $R(\boldsymbol{m})$. For random, discrete input this rule would then be a form of stochastic gradient ascent.

With this model, we observe that the objective function can be rewritten in tensor notation. Note that this input model can be seen as a kind of degenerate mixture model.

This objective function can be written as a tensor objective function, by noting the following:

$$T = \sum_k \alpha_k \boldsymbol{d}_k \otimes \boldsymbol{d}_k \otimes \boldsymbol{d}_k$$

$$M = \sum_k \alpha_k \boldsymbol{d}_k \otimes \boldsymbol{d}_k$$

$$R(\boldsymbol{m}) = \frac{1}{3} T(\boldsymbol{m}, \boldsymbol{m}, \boldsymbol{m}) - \frac{1}{4} M(\boldsymbol{m}, \boldsymbol{m})^2 \tag{2}$$

For completeness, we present a proof that the stable points of the expected BCM update are selective for only one of the data vectors.

The stable points of the expected update occur when $E[\dot{\boldsymbol{m}}] = 0$. Let $c_k = \langle \boldsymbol{m}, \boldsymbol{d}_k \rangle$ and $\phi_k = \phi(c_k, \theta)$. Let $\boldsymbol{c} = [c_1, \ldots, c_K]^T$ and $\Phi = [\phi_1, \ldots, \phi_K]^T$.

$$D^T = [\boldsymbol{d}_1 | \cdots | \boldsymbol{d}_K]$$

$$P = \operatorname{diag}(\boldsymbol{\alpha})$$

**Theorem 3.2.** *(Intrator 1992) Let $K = N$, let each $\boldsymbol{d}_k$ be linearly independent, and let $\alpha_k > 0$ and distinct. Then stable points (in the sense of Lyapunov) of the expected update $\dot{\boldsymbol{m}} = \nabla R$ occur when $\boldsymbol{c} = \alpha_k^{-1} \boldsymbol{e}_k$ or $\boldsymbol{m} = \alpha_k^{-1} D^{-1} \boldsymbol{e}_k$. $\boldsymbol{e}_k$ is the unit basis vector, so there is activity for only one stimuli.*

*Proof.* $E[\dot{\boldsymbol{m}}] = D^T P \Phi$ which is 0 only when $\Phi = 0$. Note $\theta = \sum_k \alpha_k c_k^2$. $\phi_k = 0$ if $c_k = 0$ or $c_k = \theta$. Let $S_+ = \{k : c_k \neq 0\}$, and $S_- = \{k : c_k = 0\}$. Then for all $k \in S_+$, $c_k = \beta_{S_+}$

$$\beta_{S_+} - \beta_{S_+}^2 \sum_{k \in S_+} \alpha_i = 0 \qquad\qquad \beta_{S_+} = \left( \sum_{k \in S_+} \alpha_i \right)^{-1}$$

Therefore the solutions of the BCM learning rule are $\boldsymbol{c} = \mathbb{1}_{S_+} \beta_{S_+}$, for all subsets $S_+ \subset \{1, \ldots, K\}$. We now need to check which solutions are stable. The stable points (in the sense of Lyapunov) are points where the matrix

$$H = \frac{\partial E[\dot{\boldsymbol{m}}]}{\partial \boldsymbol{m}} = D^T P \left( \frac{\partial \Phi}{\partial \boldsymbol{c}} \right) \frac{\partial \boldsymbol{c}}{\partial \boldsymbol{m}} = D^T P \left( \frac{\partial \Phi}{\partial \boldsymbol{c}} \right) D$$

is negative semidefinite.

Let $S$ be an index set $S \subset \{1, \ldots, n\}$. We will use the following notation for the diagonal matrix $I_S$:

$$(I_S)_{ii} = \begin{cases} 1 & i \in S \\ 0 & i \notin S \end{cases} \tag{3}$$

So $I_S + I_{S^c} = I$, and $\boldsymbol{e}_i \boldsymbol{e}_i^T = I_{\{i\}}$

a quick calculation shows

$$\left( \frac{\partial \phi_i}{\partial c_j} \right) = \beta_{S_+} I_{S_+} - \beta_{S_+} I_{S_-} - 2\beta_{S_+}^2 \operatorname{diag}(\boldsymbol{\alpha}) \mathbb{1}_{S_+} \mathbb{1}_{S_+}^T$$

This is negative semidefinite iff $A = I_{S_+} - 2\beta_{S_+} \operatorname{diag}(\boldsymbol{\alpha}) \mathbb{1}_{S_+} \mathbb{1}_{S_+}^T$ is negative semidefinite.

Assuming a non-degeneracy of the probabilities $\boldsymbol{\alpha}$, and assume $|S_+| > 1$. Let $j = \arg\min_{k \in S_+} \alpha_k$. Then $\beta_{S_+} \alpha_j < \frac{1}{2}$ so $A$ is not negative semi-definite. However, if $|S_+| = 1$ then $A = -I_{S_+}$ so the stable points occur when $\boldsymbol{c} = \frac{1}{\alpha_i} \boldsymbol{e}_i$ $\qquad\qquad \square$

The triplet version of BCM can be viewed as a modification of the classical BCM rule which allows it to converge in the presence of zero-mean noise. This indicates that the stable solutions of this learning rule are selective for only one data vector, $\boldsymbol{d}_k$.

Building off of the work of [2] we will use this characterization of the objective function to build a triplet BCM update rule which will converge for general mixtures, not just discrete data points.

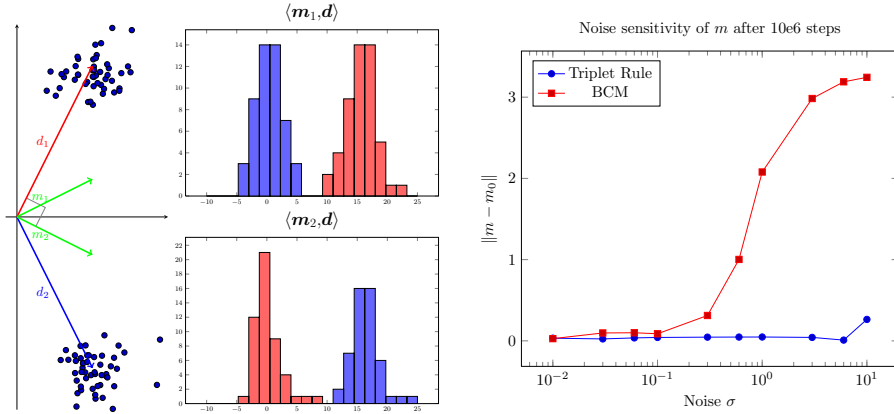

(a) Geometry of stable solutions. Each stable solution is selective in expectation for a single mixture. Note that the classical BCM rule will not converge to these values in the presence of noise.

(b) Noise response of triplet BCM update rule vs BCM update. Input data was a mixture of Gaussians with standard deviation $\sigma$. The selectivity of the triplet BCM rule remains unchanged in the presence of noise.

## 4 Triplet BCM Rule

We now show that by modifying the update rule to incorporate information from triplets of input vectors, the generality of the input data can be dramatically increased. Our new BCM rule will learn selectivity for arbitrary mixture distributions, and learn weights which in expectation are selective for only one mixture component. Assume that

$$P(\boldsymbol{d}) = \sum_k \alpha_k P_k(\boldsymbol{d})$$

where $E_{P_k}[\boldsymbol{d}] = \boldsymbol{d}_k$. For example, the data could be a mixture of axis-aligned Gaussians, a mixture of independent Poisson variables, or mixtures of independent Bernoulli random variables to name a few. We also require $E_{P_k}[\|\boldsymbol{d}\|^2] < \infty$. We emphasize that we do not require our data to come from any parametric distribution.

We interpret $k$ to be a latent variable that signals the hidden cause of the underlying input distribution, with distribution $P_k$. Critically, we assume that the hidden variable $k$ changes slowly compared to the inter-spike period of the neuron. In particular, we need at least 3 samples from each $P_k$. This corresponds to the multi-view assumption of [2]. A particularly relevant model meeting this assumption is that of spike counts in disjoint intervals under a Poisson process, with a discrete, time varying rate parameter. For the purpose of this paper, we assume the number of mixed distributions, $k$, is equal to the number of dimensions, $n$, however it is possible to relax this to $k < n$.

Let $\{\boldsymbol{d}^1, \boldsymbol{d}^2, \boldsymbol{d}^3\}$ be a triplet of independent copies from some $P_k(\boldsymbol{d})$, i.e. each are drawn from the same mixture. It is critical to note that if $\{\boldsymbol{d}^1, \boldsymbol{d}^2, \boldsymbol{d}^3\}$ are not drawn from the same class, this update will not converge to the global maximum. Numerical experiments show this assumption can be violated somewhat without severe changes to the fixed points of the algorithm. Our sample is then a sequence of triplets, each triplet drawn from the same latent distribution. Let $c^i = \langle \boldsymbol{d}^i, \boldsymbol{m} \rangle$. With these independent triples, we note that the tensors $T$ and $M$ from equation (2) can be written as moments of the independent triplets

$$T = E[\boldsymbol{d}^1 \otimes \boldsymbol{d}^2 \otimes \boldsymbol{d}^3]$$
$$M = E[\boldsymbol{d}^1 \otimes \boldsymbol{d}^2]$$
$$R(\boldsymbol{m}) = \frac{1}{3} T(\boldsymbol{m}, \boldsymbol{m}, \boldsymbol{m}) - \frac{1}{4} M(\boldsymbol{m}, \boldsymbol{m})^2$$

This is precisely the same objective function optimized by the classical BCM update, with the conditional means of the mixture distributions taking the place of discrete data points. With access to independent triplets, selectivity for significantly richer input distributions can be learned.

As with classical BCM, we can perform gradient ascent in this objective function which leads to the expected update

$$E[\nabla R] = E[c^1 c^2 \boldsymbol{d}^3 + (c^1 \boldsymbol{d}^2 + c^2 \boldsymbol{d}^1)(c^3 - 2\theta)]$$

where $\theta = E[c^1 c^2]$. This update is rather complicated, and couples pre and post synaptic firing rates across multiple time intervals. Since each $c^i$ and $\boldsymbol{d}^i$ are identically distributed, this expectation is equal to

$$E[c^2(c^3 - \theta)\boldsymbol{d}^1]$$

which suggests a much simpler update. This ordering was chosen to match the spike timing dependency of synaptic modification. This update depends on the presynaptic input, and the postsynaptic excitation in two disjoint time periods.

**Definition 4.1** (Full-rank Triplet BCM)**.** The full-rank Triplet BCM update rule is:

$$\boldsymbol{m}_t = \pi(\boldsymbol{m}_{t-1} + \gamma_t \phi(c^2, c^3, \theta_{t-1})\boldsymbol{d}^1) \tag{4}$$

where $\phi(c^2, c^3, \theta) = c^2(c^3 - \theta)$, the step size $\gamma_t$ obeys $\sum_t \gamma_t \to \infty$, and $\sum_t \gamma_t^2 < \infty$. $\pi$ is a projection into an arbitrary large compact ball, which is needed for technical reasons to guarantee convergence.

## 5  Stochastic Approximation

Having found the stable points of the *expected* update for BCM and triplet BCM, we now turn to a proof of convergence for the stochastic update generated by mixture models. For this, we turn to results from the theory of stochastic approximation.

We will decompose our update into two parts, the expected update, and the (random) deviation. This deviation will be a $L_2$ bounded martingale, while the expected update will be a ODE with the previously calculated stable points. Since the expected update is the gradient of a objective function $R$, the Lyapunov functions required for the stability analysis are simply this objective function.

The decomposition of the triplet BCM stochastic process is as follows:

$$
\begin{aligned}
\boldsymbol{m}_t - \boldsymbol{m}_{t-1} &= \gamma_t \phi(c_t^2, c_t^3, \theta_{t-1})\boldsymbol{d}^1 \\
&= \gamma_n E[\phi(c^2, c^3, \theta_{t-1})\boldsymbol{d}^1] + \gamma_n \left(\phi(c^2, c^3, \theta_{t-1})\boldsymbol{d}^1 - E[\phi(c^2, c^3, \theta_{t-1})\boldsymbol{d}^1]\right) \\
&= \gamma_t h(\boldsymbol{m}_t) - \gamma_t \eta_t
\end{aligned}
$$

Here, $h(\boldsymbol{m}_t)$ is the deterministic expected update, and $\eta_t$ is a martingale. All our expectations are taken with respect to triplets of input data. The decomposition for classical BCM is similar.

This is the Doob decomposition [8] of the sequence. Using a theorem of Delyon [7], we will show that our triplet BCM algorithm will converge with probability 1 to the stable points of the expected update. As was shown previously, these stable points are selective for one and only one mixture component in expectation.

**Theorem 5.1.** *For the full rank case, the projected update converges w.p. 1 to the zeros of $\nabla \Phi$*

*Proof.* See supplementary material, or an extended discussion in a forthcoming arXiv preprint [13]. □

## 6  Triplet BCM Explains STDP Up to Spike Triplets

Biophysically synaptic efficiency in the brain is more closely modeled by spike timing dependent plasticity (STDP). It depends precisely on the interval between pre- and post-synaptic spikes. Initial research on spike pairs [15, 3] showed that a presynaptic spike followed in close succession by a postsynaptic spike tended to strengthen a synapse, while the reverse timing weakened it. Later work on natural spike chains [9], triplets of spikes [4, 19], and quadruplets have shown interaction effects beyond pairs. Most closely to ours, recent work by Pfister and Gerstner [17] suggested that a synaptic modification function depending only on spike triplets is sufficient to explain all current experimental data. Furthermore, their rule resembles a BCM learning rule when the pre- and post-synaptic firing distributions are independent Poisson.

We now demonstrate that our learning rule can model both the pairwise and triplet results from Pfister and Gerstner using a smaller number of free parameters and without the introduction of hidden leaky timing variables. Instead, we work directly with the pre- and post-synaptic voltages, and model the natural voltage decay during the falling phase of an action potential. Our (four) free variables are the voltage decay, which we set within reasonable biological limits; a bin width, controlling the distance between spiking triplet periods; $\theta$, our sliding voltage threshold; and an overall multiplicative constant. We emphasize that our model was not designed to fit these data; it was designed to learn selectivity for the multi-view mixture task. Spike timing dependence falls out as a natural consequence of our multi-view assumption.

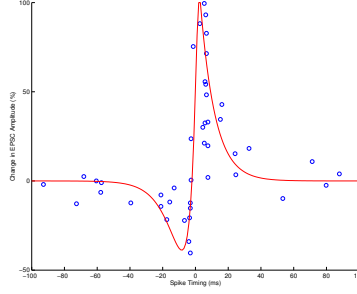

Figure 2: Fit of triplet BCM learning rule to synaptic strength STDP curve from [3]. Data points were recreated from [3] . Spike timing measures the time between post synaptic and presynaptic spikes, $t_{post} - t_{pre}$. A positive time means the presynaptic spike was followed by a postsynaptic spike.

We first model hippocampus data from Mu-ming Poo [3], who applied repeated electrical stimulation to the pre- and post-synaptic neurons in a pairing protocol within which the relative timing of the two spike chains was varied. After repeated stimulation at a fixed timing offset, the change in synaptic strength (postsynaptic current) was measured.

We take the average voltage in triplet intervals to be the measure of pre- and post-synaptic activity, and consider a one-dimensional version of our synaptic update:

$$\delta m = Ac^2(c^3 - \theta)d^1 \tag{5}$$

where $c^2$ and $c^3$ are the postsynaptic voltage averaged over the second and third time bins, and $d^1$ is the presynaptic voltage averaged over the first time bin. We assume our pre and post synaptic voltages are governed by the differential equation:

$$\frac{dV}{dt} = -\tau V \tag{6}$$

such that, if $t = s_k$ where $s_k$ is the $k$th spike, $V(t) \to 1$. That is, the voltage is set to 1 at each spike time before decaying again.

Let $V_{pre}$ be the presynaptic voltage trace, and $V_{post}$ be the postsynaptic voltage trace. They are determined by the timing of pre- and post-synaptic spikes, which occur at $r_1, r_2, \ldots, r_n$ for the presynaptic spikes, and $o_1, o_2, \ldots o_m$ for the postsynaptic spikes.

To model the pairwise experiments, we let $r_i = r_0 + iT$ where $T = 1000$ms, a large time constant. Then $o_i = r_i + \delta_t$ where $\delta_t$ is the spike timing. Let $\delta_b$ be the size of the bins. That is to say,

$$d^1(t) = \int_{t-\frac{\delta_b}{2}}^{t+\frac{\delta_b}{2}} V_{pre}(t' + \delta_b)dt' \qquad c^2(t) = \int_{t-\frac{\delta_b}{2}}^{t+\frac{\delta_b}{2}} V_{post}(t')dt'$$

$$c^3(t) = \int_{t-\frac{\delta_b}{2}}^{t+\frac{\delta_b}{2}} V_{post}(t' - \delta_b)dt' \qquad V_{post}(t) = V_{pre}(t - \delta_t)$$

Then the overall synaptic modification is given by

$$\int_t Ac^2(t)(c^3(t) - \theta)d^1(t)dt$$

We fit $A$, $\tau$, $\theta$, and the bin size of integration. Recall that the sliding threshold, $\theta$ is a function of the expected firing rate of the neuron. Therefore we would not expect it to be a fixed constant. Instead, it should vary slowly over a time period much longer than the data sampling period. For the purpose of these experiments it would be at an unknown level that depends on the history of neural activity. See figure 2 for the fit for Mu-ming Poo's synaptic modification data.

Froemke and Dan also investigated higher order spike chains, and found that two spikes in short succession did not simply multiply in their effects. This would be the expected result if the spike timing dependence treated each pair in a triplet as an independent event. Instead, they found that a presynaptic spike followed by two postsynaptic spikes resulted in significantly less excitation than expected if the two pairs were treated as independent events. They posited that repeated spikes interacted suppressively, and fit a model based on that hypothesis. They performed two triplet experiments with pre- pre-post triplets, and pre-post-post triplets. Results of their experiment along with the predictions based on our model are presented in figure 3.

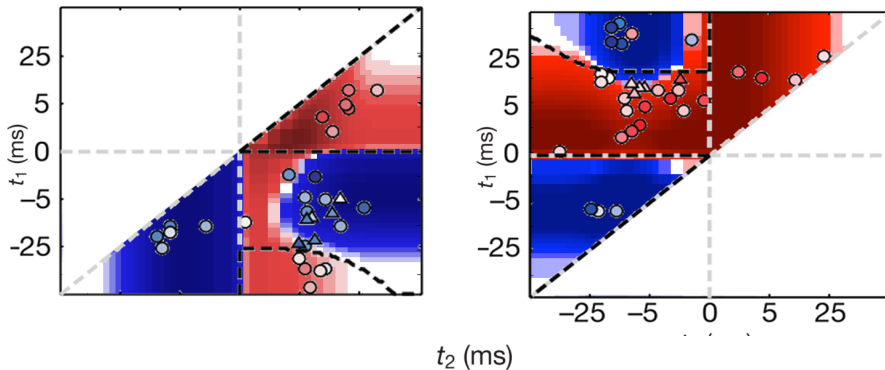

Figure 3: Measured excitation and inhibition for spike triplets from Froemke and Dan are demarcated in circles and triangles. A red circle or triangle indicates excitation, while a blue circle or triangle indicates inhibition. The predicted results from our model are indicated by the background color. Numerical results for our model, with boundaries for the Froemke and Dan model are reproduced.

Left figure is pairs of presynaptic spikes, and a single post-synaptic spike. The right figure is pairs of postsynaptic spikes, and a presynaptic spike. For each figure, $t_1$ measures the time between the first paired spike with the singleton spike, with the convention that each $t$ is positive if the presynaptic spike happens before the post synaptic spike. See paired STDP experiments for our spiking model. For the top figure, $\theta = .65$, our bin width was 2ms, and our spike voltage decay rate $\tau = 8$ms. For the right figure $\theta = .45$. Red is excitatory, blue is inhibitory, white is no modification. A positive $t$ indicates a presynaptic spike occurred before a postsynaptic spike.

## 7   Conclusion

We introduced a modified formulation of the classical BCM neural update rule. This update rule drives the synaptic weights toward the components of a tensor decomposition of the input data. By further modifying the update to incorporate information from triplets of input data, this tensor decomposition can learn the mixture means for a broad class of mixture distributions. Unlike other methods to fit mixture models, we incorporate a multiview assumption that allows us to learn asymptotically *exact* mixture means, rather than local maxima of a similarity measure. This is in stark contrast to EM and other gradient ascent based methods, which have limited guarantees about the quality of their results. Conceptually our model suggests a different view of spike waves during adjacent time epochs: they provide multiple independent samples of the presynaptic "image."

Due to size constraints, this abstract has has skipped some details, particularly in the experimental sections. More detailed explanations will be provided in future publications.

Research supported by NSF, NIH, The Paul Allen Foundation, and The Simons Foundation.

## Footnotes

[1]Now at Google Inc.

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
