[Supplementary Material]

# 1 Proof of Theorem 1

**Definition 1.1.** (Delyon 1996)

Recall our update

$$\boldsymbol{m}_n - \boldsymbol{m}_{n-1} = \gamma_n h(\boldsymbol{m}_n) - \gamma_n \eta_n$$

Let $\gamma_n$ be a sequence with $\sum_{i=0}^{\infty} \gamma_i = \infty$ and $\sum_{i=0}^{\infty} \gamma_i^2 < \infty$. Let $\eta_n$ be a perturbation, $\eta_n = e_n + r_n$. A stochastic algorithm is *A-stable* if $\boldsymbol{m}_n \in K_0$ infinitely often, where $K_0$ is a compact subset of $\mathbb{R}^n$ and the series $\sum \gamma_n e_n$ or $\sum \gamma_n e_n \mathbb{1}_{V(\boldsymbol{m}_n) \leq M}$ converges for all $M$ and $r_n \to 0$.

For technical reasons, we project our weights down to a reasonable compact set where we know the true parameters lie if they ever become unreasonably large. We note that this set can be made arbitrarily large, and for a sufficiently small initial step size we have found this projection does not need to be done in practice. This ensures that the sequence returns infinitely often to a compact set. We note that biological neurons also have physical limitations on their selectivity, which act as effective projections.

**Theorem 1.2.** *(Delyon 1996) The vector field $h$ is defined on an open set $\mathcal{O} \subset \mathbb{R}$. There exists a nonnegative $C_1$ Lyapunov function $V$ and a finite set $\mathcal{K} \subset \mathcal{O}$ s.t.*

*1) $V(x)$ tends to $+\infty$ if $x \to \partial\mathcal{O}$ or $|x| \to \infty$*

*2) $h$ is continuous and $\langle \nabla V(x), h(x) \rangle < 0$ if $x \notin \mathcal{K}$*

*3) Conditions for Projection: Let $\pi(x)$ be a continuous projection onto a compact set $\mathcal{Q} \subset \mathcal{O}$ s.t. $\pi(x) = x$ for $x \in \mathcal{Q}$, and $\langle \nabla V(x), \pi(x) - x \rangle < -\delta|\pi(x) - x|$ for some $x$ in $\mathcal{O} \backslash \mathcal{Q}$*

*Let $\boldsymbol{m}_n = \boldsymbol{m}_{n-1} + \gamma_n h(\boldsymbol{m}_{n-1}) + \gamma_n \eta_n$ We further require that the stochastic algorithm is A-stable. Then, $d(\boldsymbol{m}_n, \mathcal{K})$ converges to $0$.*

**Theorem 1.3.** *For the full rank case, the projected update converges w.p. 1 to the zeros of $\nabla\Phi$*

*Proof.* Let $\mathcal{O}$ be an open neighborhood of $B$. We replace our update with its projected version

$$\boldsymbol{m} = \pi(\gamma_n \phi(c^2, c^3, \theta_{n-1})\boldsymbol{d}^1) \tag{1}$$

This projection gives us the first part of the A-stability immediately. Furthermore, the bounded variance of each $P_k$ and the boundedness of $\boldsymbol{m}$ means each $c$ has bounded variance, so the martingale increment has bounded variance. This, plus the requirement that $\sum \gamma_i^2 < \infty$ means the martingale is bounded in $L_2$ so it converges. This gives us the A-stability of the sequence.

Let $V = -R$ then conditions 1) and 2) of Delyon are clearly satisfied. The optional projection requirement is satisfied by noting that for some $C$

$$\frac{1}{C}\boldsymbol{m}^T M \boldsymbol{m} < \|\boldsymbol{m}\|^2 < C\boldsymbol{m}^T M \boldsymbol{m}$$

and for large enough $\boldsymbol{m}$

$$\langle \nabla \Phi, \pi(\boldsymbol{m}) - \boldsymbol{m} \rangle < C \|\boldsymbol{m}\|^4))$$
$$\text{and } \|\pi(\boldsymbol{m} - \boldsymbol{m})\| = C'(O(\|\boldsymbol{m}\|))$$

where $C' = \frac{r}{\boldsymbol{m}^T \boldsymbol{m}} - 1$ so for sufficiently large $r$ the optional projection requirement is satisfied. Therefore the stochastic algorithm converges with probability 1 to the zeros of $\nabla R$. $\qquad\square$