[Reviews · NeurIPS 2014]

Submitted by Assigned_Reviewer_4

Summary:
The paper extends the classical BCM learning rule to utilize information from spike triplets.
It is shown that the update rule can learn selectivity for mixture distributions (by converging
to the class means).

Quality:
By employing tensor notation, the paper shows that the BCM rule can be generalized to use information from
more than a pair of spikes (spike triplets are used for the examples). While the model has fewer parameters
than previous learning algorithms based on spike triplets or quadruplets and the method can be shown to have
stable points as class means of mixture distributions, the lack of experimental comparisons with other models
makes it hard to gauge the incremental contribution of the model.

Clarity:
The paper is in general easy to follow but the derivation of equations on line 269 and 273 was unclear to me.

Originality:
STDP utilizing spike triplets has been introduced in the past by Pfister and Gerstner. Also, Froemke and Dan
published a model which uses spike quadruplets. Moreover, it seems to me that under Poisson spiking statistics
and certain choice of parameters, the model from Pfister and Gerstner is equivalent to the model described here.
In that light, I believe a clear comparison should be made using simulations of the algorithm with those from
Pfister/Gertsner, Froemke, etc. The simulations should clearly demonstrate under what circumstances does the
presented model differ (in the expected sense) and is better than that from Pfister and Gerstner.

Significance:
The paper extends a classical approach in neural learning (BCM) and demonstrates that the extended approach has broader
capabilities than the original approach. However, clear and detailed quantitative comparison to previous research in the field is missing.

[1] Pfister, Jean-Pascal, and Wulfram Gerstner. "Triplets of spikes in a model of spike timing-dependent plasticity." The Journal of neuroscience 26.38 (2006): 9673-9682.

Notes:

Line 79, Baysian -> Bayesian
Line 99 \delta_{ij} is the more common notation rather than \delta_i^j
Line 106 In the definition of c, subscript i should be added.
Does the example here have multiple neurons or a single neuron and dimensional data?
If the former, m_i, m_{i-1} and d_i should be bold in the equaution. If not, c_i needs
to be defined as m_{i-1} x d_i (without inner product and bold letters) to clarify its
a single neuron.
Line 147, 148 I think there should be an expectation in front RHS equations on both lines
Line 161 m should be bold in R(m)
Line 262 same as above
Line 351 Missing verb
Line 410 "Lower figure" needs to be corrected. The figures are next to each other.
Summary: While the extension in the paper is analytically shown to have broader capabilities than the classical BCM rule,
it is not clear how this model compares in performance to previous learning models employing more than a
pair of spikes. Also under certain commonly made assumptions, the model seems to be equivalent to a previously
proposed model.

Submitted by Assigned_Reviewer_24

The paper discusses an online neural architecture to estimate the means of mixtures.
The author(s) proposed a method that is an extension of the classical BCM rule, which is a corrected algorithm of the Hebb learning and formulated using tensor decomposition. The method assumes that three-time measurement is possible while the stimulus remains constant and calculates the means of mixture in a stochastic-approximation manner.
The author(s) also showed the algorithm is related to STDP, which implies the significance of the results in neuroscience.

[quality]
The idea of formulating this estimation problem as tensor decomposition is interesting. The paper gave a new insight to learning of spiking neurons.

[clarity]
The paper is well written and rather easy to follow.

[originality]
This work seems highly original.

[significance]
The proposed algorithm can be implemented in neural systems using SDTP and gives a new insight to computational neuroscience.
Summary: The idea of formulating the estimation of the means of mixtures as tensor decomposition is interesting. The paper also showed the algorithm can be implemented as SDTP, which gives an insight to computational neuroscience.

Submitted by Assigned_Reviewer_42

A reformulation of the Bienenstock-Cooper-Munroe (BCM) learning rule using tensors is introduced. It is used to derive an algorithm based on triplets of input samples. The algorithm performs a stochastic gradient ascent on the same objective function as the BCM rule. It is claimed that far 'richer distributions' than Poisson can be learned, and proven that the output neurons will become selective for only one of the data vectors. Finally, the synaptic changes that would be induced by triplets of spikes are discussed. While the authors use terms that are not appropriate in this context (e.g. they call the synaptic traces caused after presynaptic and postsynaptic action potentials 'voltages') the results nicely match the findings of Froemke and Dan (their refererence [9]). I found the paper hard to follow partly because some details are missing. However, if correct the paper is very interesting, in particular because it sheds a fresh light on the relation of regulatory mechanisms as in the BCM-rule and spike-timing dependent plasticity.
Summary: A reformulation of the Bienenstock-Cooper-Munroe (BCM) learning rule using tensors is introduced. While quite formal the paper is interesting because it sheds a fresh light on the relation of regulatory mechanisms as in the BCM-rule and spike-timing dependent plasticity.
Author Feedback
Author rebuttal: We thank the reviewers for their thoughtful reading of our manuscript, and for the positive feedback from all of them. Our major contributions -- the tensor formulation of BCM learning and the stochastic convergence proof -- are technical and difficult to compress into this short paper. Of course one needs the full proofs and, should the paper be accepted, we plan to have arXiv preprints referenced in the Proceedings. We also thank the second reviewer for his suggestions on notation, which would be tightened to highlight the distinctions between our model and classical treatments of BCM.

We agree with the second reviewer that there is an important relationship to the triplet rule of Pfister and that, for certain parameters in their rule, it is equivalent to ours. (This was discussed briefly in Sec. 6 of our paper for both the Pfister and the Dan rules.) We stress that the Pfister rule was essentially developed by empirically
fitting a third-order Volterra expansion, and a theoretical justification was given only for orthogonal Poisson input. Our rule follows naturally from a theoretical solution to a much richer and more natural learning problem.

To quote a more recent paper [1] from the Pfister group:

"The general problem of deriving selectivity analytically has not yet been solved; however, numerical simulations suggest that the triplet rule successfully drives selectivity even when the rate-based inputs are nonorthogonal.''

We believe the analytical solution and extension to non-Poisson and non orthogonal input is our main contribution, plus the convergence proof.

[1] A triplet spike-timing–dependent plasticity model generalizes the Bienenstock–Cooper–Munro rule to higher-order spatiotemporal correlations
www.pnas.org/cgi/doi/10.1073/pnas.1105933108